# Oral Exposure to Titanium Dioxide E171 and Zinc Oxide Nanoparticles Induces Multi-Organ Damage in Rats: Role of Ceramide

**DOI:** 10.3390/ijms25115881

**Published:** 2024-05-28

**Authors:** Rocío Bautista-Pérez, Agustina Cano-Martínez, Manuel Alejandro Herrera-Rodríguez, María del Pilar Ramos-Godinez, Olga Lidia Pérez Reyes, Yolanda Irasema Chirino, Zariá José Rodríguez Serrano, Rebeca López-Marure

**Affiliations:** 1Departamento de Biología Molecular, Instituto Nacional de Cardiología Ignacio Chávez, Mexico City 14080, Mexico; maría.bautista@cardiologia.org.mx; 2Departamento de Fisiología, Instituto Nacional de Cardiología Ignacio Chávez, Mexico City 14080, Mexico; agustina.cano@cardiologia.org.mx (A.C.-M.);; 3Departamento de Microscopía Electrónica, Instituto Nacional de Cancerología, Mexico City 14080, Mexico; 4Departamento de Patología, Instituto Nacional de Cardiología Ignacio Chávez, Mexico City 14080, Mexico; 5Unidad de Biomedicina, Facultad de Estudios Superiores Iztacala, Universidad Nacional Autónoma de México, Mexico City 54090, Mexico

**Keywords:** titanium dioxide nanoparticles, zinc oxide nanoparticles, morphological changes, organ injury, ceramide, peroxynitrite, lysosome membrane permeabilization

## Abstract

Food-grade titanium dioxide (E171) and zinc oxide nanoparticles (ZnO NPs) are common food additives for human consumption. We examined multi-organ toxicity of both compounds on Wistar rats orally exposed for 90 days. Rats were divided into three groups: (1) control (saline solution), (2) E171-exposed, and (3) ZnO NPs-exposed. Histological examination was performed with hematoxylin–eosin (HE) staining and transmission electron microscopy (TEM). Ceramide (Cer), 3-nitrotyrosine (NT), and lysosome-associated membrane protein 2 (LAMP-2) were detected by immunofluorescence. Relevant histological changes were observed: disorganization, inflammatory cell infiltration, and mitochondrial damage. Increased levels of Cer, NT, and LAMP-2 were observed in the liver, kidney, and brain of E171- and ZnO NPs-exposed rats, and in rat hearts exposed to ZnO NPs. E171 up-regulated Cer and NT levels in the aorta and heart, while ZnO NPs up-regulated them in the aorta. Both NPs increased LAMP-2 expression in the intestine. In conclusion, chronic oral exposure to metallic NPs causes multi-organ injury, reflecting how these food additives pose a threat to human health. Our results suggest how complex interplay between ROS, Cer, LAMP-2, and NT may modulate organ function during NP damage.

## 1. Introduction

Metallic nanoparticles (NPs) such as gold (Au), silver (Ag), copper (Cu), iron (Fe), zinc (Zn), and titanium (Ti) and their oxides have multiple applications in food packaging and storage as well as enhancing organoleptic properties (taste, color, texture) [1]. NPs increase safety, preserve nutritional value, and extend product shelf life [2]. Other NPs applications include drug delivery systems and antimicrobial treatment [3]. NPs are widely used in biomedicine, being used directly in patients [4]. These applications have increased human exposure to NPs and their oxides such as aluminum oxide (Al_2_O_3_), cerium oxide (CeO_2_), titanium dioxide (TiO_2_), and zinc oxide (ZnO) [5].

Apparently, these bulk materials have no adverse effects on health; however, NPs can cross physiological barriers and consequently enter the bloodstream. TiO_2_ NPs and ZnO NPs have been detected in blood, tissues, and several organs including the brain, heart, intestine, kidney, liver, lung, and stomach of mice and rats after inhalation, intraperitoneal, or oral exposure [6,7,8,9,10,11,12]. Exposure to TiO_2_ NPs and ZnO NPs can cause vascular dysfunction in subcutaneous arteries, coronary arteries, and the aorta [13,14]; histological and functional changes in organs [15,16]; genotoxicity [17]; cytotoxicity [18,19]; inflammation [20,21]; and endoplasmic reticulum (ER) stress [22,23]. Depending on dose and physicochemical characteristics (components, charges, solubility, size, shape), TiO_2_ NPs and ZnO NPs produce reactive oxygen species (ROS), including radical and non-radical species such as peroxynitrite, activating different signaling pathways, eventually altering multiple cellular processes including mitosis, apoptosis, or autophagy [24,25].

Interestingly, TiO_2_ NPs promote ROS, apoptosis, and lysosome membrane permeabilization (LMP) in HaCaT cells [26]. Lysosomal membrane stability is a key factor determining cell survival–death signaling. Full LMP apparently favors a necrosis onset, whereas partial LMP triggers apoptosis [27,28]. The lysosomal membrane is protected from acidic hydrolases by highly glycosylated membrane-bound proteins such as LAMP-1/2. The down-regulated expression of these glycoproteins increases susceptibility to LMP [29].

Moreover, exposure of pulmonary epithelial cells to TiO_2_ NPs leads to autophagy, apoptosis, and increased ceramide levels [30]. Cross-talk between Cer and ROS is proposed as a critical event for cell death signaling [31]. On the other hand, tyrosine, a non-essential amino acid, is often surface-exposed in proteins, therefore becoming modified by nitration caused by nitric oxide (NO) formation to produce NT [32]. Increased NT levels correlate with elevated levels of other oxidative stress indicators [33]. These findings suggest how complex interplay between ROS, Cer, LMP, and NT may modulate organ functions during damage. Since NPs exposure as food additives has exponentially increased and becomes oxidative stress, a key mechanism associated with NPs damage, we explored E171 and ZnO NPs effects in histology, cellular morphology, and organ damage markers such as Cer, NT, and LAMP-2 in chronically exposed rats.

## 2. Results

### 2.1. Organ Histology and Cell Morphology

Since NPs from food additives enter the bloodstream, affecting multiple organs, we evaluated organ histology through HE staining and TEM, respectively.

Rat aortas exposed to E171 and ZnO NPs exhibited disorganized vascular walls, aneurysms (thick arrows), and cellular infiltration (thin arrows) (Figure 1A). The TEM analysis of controls showed unaltered endothelial and smooth muscle cells as well as intact nuclei (Figure 1B). Aortas from rats exposed to both E171 and ZnO NPs displayed vacuoles and intercellular spaces. Furthermore, aortic adventitia from E171-exposed rats revealed an increased number and size of adipocytes.

Brain HE-stained slides from E171-exposed rats (thin arrows) and those from ZnO NPs-exposed ones (thick arrows) displayed “holes” typical of aneurysm formation, not observed in controls (Figure 2A). TEM of control brains showed well-preserved chromatin, intermediate filaments, nuclei, nucleoli, and mitochondria in astrocytes (Figure 2B). Brains of E171-exposed rats presented astrocytes with brighter cytoplasm, eccentric nucleoli, and less chromatin. On the other hand, brains from ZnO NPs-exposed rats exhibited less cristae, indicating mitochondrial dysfunction and brain damage.

HE staining of myocardial tissue showed that E171 and ZnO NPs produced broken fibers and cellular infiltration (arrows) compared to controls (Figure 3A). The TEM analysis of control rat hearts showed ordered mitochondria with well-defined and conserved cristae, and well-structured sarcomeres (Figure 3B). However, in E171-exposed rats, loss of sarcomere continuity and mitochondrial damage are evident. Hearts of ZnO NPs-exposed rats showed greater damage, fragmented sarcomeres with large electro-lucid spaces (highlighting edema, indicated by arrows), and disorganized mitochondria with structural damage.

E171 and ZnO NPs exposure promoted structural changes in the small intestine wall observed through HE staining. Exposed rats exhibited discontinuity and disorganization of intestinal villi, disorganized mucosa, and a less compact submucosal layer with elongated fibers compared to controls (Figure 4A). TEM of control intestines showed goblet cells with abundant microvilli of a uniform size (Figure 4B). Intestines of E171-exposed rats showed intercellular electro-lucent vacuoles, indicating edema. Intestines of ZnO NPs-exposed rats showed no ultrastructural changes; however, goblet cells were present.

HE staining showed that E171 and ZnO NPs caused structural alterations at the glomerular level, i.e., the disorganization of capillaries, podocytes, and mesangial cells; elongated and thin Bowman’s capsule wall; and increased Bowman’s space (arrows). An erythrocyte presence indicated interstitial hemorrhage between proximal tubules (Figure 5A). TEM showed tubular epithelial cells with intact microvilli and numerous mitochondria in control rat kidneys (Figure 5B). Kidneys of E171-exposed rats exhibited mitochondrial deterioration with cristae loss and expanded ER. ZnO NPs exposure promoted ultrastructural damage in mitochondria including cristae loss, modified microvilli, and edema.

Liver HE staining revealed that both NPs caused the disorganization of hepatic tissue, cellular infiltration (arrows), and hemorrhage (interstitial erythrocytes in brown) (Figure 6A). TEM revealed nuclei with abundant euchromatin and eccentric nucleoli in hepatocytes from control rats, and typical mitochondria, as well as normal and abundant rough and smooth ER (Figure 6B). In addition, we observed glycogen deposits. Conversely, livers of E171-exposed rats exhibited mitochondrial damage, dilated rough ER, and less electro-dense nuclei. Liver cells exposed to ZnO NPs displayed larger intercellular spaces, denoting edema, and weak electron density of nuclei. Smooth ER was dilated whereas few mitochondria exhibited intercellular spaces. Both NPs modified hepatic glycogen deposition; thus, alterations in ER and mitochondria may reflect metabolic changes.

### 2.2. Ceramide, Nitrotyrosine, and LAMP-2 Expression

To assess NP-derived tissue injury across different organs, biomarkers such as ceramide (Cer), nitrotyrosine (NT), and LAMP-2 (lysosomes) were detected by immunofluorescence assays.

We detected up-regulated Cer and NT expression in aortas from E171- and ZnO NPs-exposed rats. No changes in LAMP-2 expression were observed with any NPs (Figure 7). After ZnO NPs exposure, yellow-colored Cer and NT co-localized in aortic elastic sheets, mainly at the intima level (Figure 7A, merged), while in E171-exposed rats, this was not observed.

Immunofluorescence quantification showed 5- and 4.5-fold higher Cer expression in the brain cortex from E171- and ZnO NPs-exposed rats, respectively. We found up-regulated brain NT and LAMP-2 expression after exposure to both NPs compared to controls (Figure 8).

We observed elevated Cer and NT expression in rat hearts after exposure with both NPs, while ZnO NPs exposure slightly increased LAMP-2 expression (Figure 9). Cer and NT were present in broken myocardial fibers from E171-exposed rats (yellow arrow) (merged), while ZnO NPs-exposed rats displayed signals for all three biomarkers in fragmented myocardial fibers (Cer, NT, and LAMP-2) (white arrows) (Figure 9, merged).

At the intestines, no changes in Cer and NT expression occurred after exposure with both NPs; however, E171 and ZnO NPs significantly up-regulated intestinal LAMP-2 expression (Figure 10). Furthermore, LAMP-2 co-localized with Cer at the intestinal tract (white arrows) (Figure 10, merged).

E171 and ZnO NPs (Figure 11) increased the renal expression of Cer, NT, and LAMP-2. E171 stimulated Cer, NT, and LAMP-2 expression at renal tubules, while ZnO NPs up-regulated Cer, NT, and LAMP-2 levels in glomeruli. In both exposures Cer, NT, and LAMP-2 co-localized (arrows) (Figure 11, merged).

In addition, both NPs significantly increased hepatic Cer, NT, and LAMP-2 expression (Figure 12). In this organ, ZnO NPs displayed the strongest effect. Furthermore, Cer co-localized with LAMP-2 in hepatic tissue (Figure 12, merged).

## 3. Discussion

Metallic nanoparticles are increasingly manufactured into commercial products, particularly foods [34]. Nanomaterials used as food additives, drugs, or cosmetics are regulated by the Federal Food, Drug, and Cosmetic Act (FFDCA) of the USA; however, these chemicals represent a public health risk. TiO_2_ NPs and ZnO NPs are major food additives for human use; therefore, our study tested them for multi-organ injury, produced by chronic oral administration in rats. Given their nanometric size, ingested NPs diffuse along blood vessels and contact several organs, tissues, and cell types. Accordingly, we evaluated NPs’ impact on different organs including the intestine, aorta, heart, brain, kidney, and liver. First, histological changes were studied through HE staining; then, cellular ultrastructure was observed by TEM.

Our experiments revealed structural modifications (histological lesions) in multiple organs of rats exposed to both NPs. The TEM analysis showed an increased number of vacuoles (edema), intercellular spaces, and mitochondrial alterations. HE staining displayed structural discontinuity and disorganization, hemorrhage, and inflammatory infiltration, in agreement with previous studies [35,36]. TiO_2_ NPs and ZnO NPs trigger inflammation and oxidative stress [21,37], leading to ER stress, Golgi fragmentation, and mitochondrial and lysosomal injury [12,38]. The effect of these NPs depends on the tissue type, and initial responses are mediated by a variety of cells. These include polymorphonuclear leukocytes, platelets, monocytes/macrophages, mast cells, epithelial cells, or endothelial cells [39]. Upon activation, these cells release a variety of inflammatory mediators. Examples are histamine, serotonin, eicosanoids, cytokines, ROS, NO, and Cer [40,41]. In different organs (aorta, kidney, liver, brain), NO is produced locally during inflammation by inducible nitric oxide synthase (iNOS). High NO levels combined with superoxide anions yield peroxynitrite, causing diffuse organ injury [42,43]. Our results also demonstrated elevated NT expression in several organs from rats exposed with both NPs, except intestines. NT indicates oxidative post-translational modification, affecting activity of plasma membrane proteins and cellular organelles [44]. We observed higher Cer and LAMP-2 levels as tissue damage and lysosome biomarkers in several organs after exposure with both NPs.

Cer accumulation could have different sources such as sphingomyelin hydrolysis at the plasma membrane, de novo biosynthesis at RE, and endosomes/lysosomes [45]. Mitochondrial membranes are also sources of Cer production, especially under oxidative stress and inflammation. Long-chain ceramides such as C16 form channels on the mitochondrial membrane, inducing apoptosis or necrosis [46,47]. Cer accumulates in tissues as part of the stress response to cytokines, UV light, glucocorticoids, heat shock, and toxic compounds. Thus, alterations in Cer metabolism and accumulation in different organs (kidney, heart, liver, brain, aorta) result in tissue damage and organ failure [48,49,50]. Our results show that E171 exposure produces Cer accumulation in proximal tubules, while ZnO NPs generate its accumulation in glomeruli. This could indicate renal damage due to energy demand and dependence on aerobic fatty acid oxidation necessary for active solute reabsorption, a situation in which a proximal tubule is especially vulnerable to metabolic injury. Altered lipid metabolism (characterized by mitochondrial impairment and lipid accumulation) is a common feature in various etiologies of acute kidney disease [51]. Thus, mechanisms underlying Cer effects are unclear but may involve plasma membrane modifications and/or the activation of different downstream signaling pathways.

Interestingly, our results provide evidence on Cer and NT co-localization in the aorta, heart, and kidney from rats exposed to both NPs. These data agree with previous reports on cross-talk between Cer and ROS affecting the cell fate and organ activity (endothelial, renal, glomerular, tubular, and heart function). Cer-induced endothelial dysfunction is mediated by peroxynitrite formation, resulting from increased superoxide production and reduced NO bioavailability [52,53]. After exposure to E171 and ZnO NPs, we observed cellular infiltration, the depletion of vascular smooth muscle cells, and Cer accumulation typical of aortic aneurysms [54]. This associates with increased sphingomyelin levels in bicuspid and tricuspid aortic valve aneurysms reported in previous studies, possibly through sphingomyelinase inhibition, altering the sphingomyelinase–ceramide pathway and inhibiting tissue regeneration, a potential basis for disease initiation and progression [55]. Aortic Cer and NT accumulation after exposure to ZnO NPs in this work is consistent with previous reports on aortic damage, reduced wall thickness, high NT immunoreactivity, oxidative stress, and impaired vasodilation in acetaminophen-exposed rats [56]. These mechanisms could explain how NPs cause vascular dysfunction in subcutaneous, coronary arteries and the aorta [13,14].

E171 and ZnO NP exposure produced aneurysms in rat brains. Cerebral aneurysms have pathophysiological characteristics relevant in non-traumatic brain damage, which could lead to intracranial hemorrhage (aneurysm rupture), increasing cranial pressure and oxidative stress, hypoxia/ischemia, necrosis, immune infiltration, and blood–brain barrier disruption [57]. Degenerative changes and cerebral aneurysms are associated with NO metabolism [58]. The immune response and lysosome formation increase in brain aneurysm ruptures [59]. On the other hand, E171 and ZnO NPs exposure caused lesions at intestinal walls, discontinuous epithelia, edema (electro-lucent vacuoles for E171), increased goblet cells (ZnO NPs), and LAMP-2 expression. This could affect intestinal function correlated with autophagy as a lysosome-dependent pathway [60]. Damage of small intestine walls triggers lysosome formation [61] and permeabilization, as occurring for lysosomotrophic compounds [62]. Interestingly, E171- and ZnO NPs-exposed rats showed increased liver glycogen stores. Altered glycogen metabolism could meet energy demands after tissue damage as in pathological conditions including diabetes mellitus, non-alcoholic fatty liver disease, and hepatocellular carcinoma [63].

We found several indicators of cell membrane damage and alterations of the endomembranous system in NPs-exposed rats. These include Cer accumulation, LAMP-2 and NT up-regulation, brain aneurysms, mitochondrial damage (cerebral cortex, heart, kidney, liver), cellular infiltrates (heart, liver, kidney), increased Bowman’s space, interstitial hemorrhage between proximal tubules (kidney), and dilated rough ER (liver) [47]. TiO_2_ NPs have been detected in liver and kidney lysosomes [64,65]. Lysosomes degrade macromolecules internalized by endocytosis, phagocytosis, and autophagy. LAMPs are required for lysosomal mobility, chaperone-mediated autophagy, and lysosomal fusion with autophagosomes/phagosomes to form autolysosomes/phagolysosomes, which degrade internalized material, to maintain cell homeostasis and organelle renewal. Also, LAMPs protect lysosomal membrane integrity and lacking them results in LMP. The reduction in mature lysosomes induced by LMP inhibits autophagy-dependent protein degradation [66,67]. LMP is a type of lysosomal dysfunction associated with NPs. LMP represents a cell death process resulting in the permeabilization of the mitochondrial outer membrane through several mechanisms, including lysosomal iron-mediated oxidative stress, cathepsin release, and other lysosomal hydrolases. Mitochondrial permeabilization resulting from partial LMP generates ROS and apoptosis, while massive LMP promotes cytosolic acidification and necrosis [68,69,70]. Here, we found up-regulated LAMP-2 expression across organs exposed to both NPs, except the aorta. This glycoprotein increases due to the accumulation and storage of not-digestible materials in lysosomes as previously observed [71,72]. Thus, alteration in lysosomal–sphingolipid metabolism may lead to lysosome dysfunction. This may explain our observations on Cer and LAMP-2 co-localization in livers and intestines exposed to ZnO NPs.

Particle size can affect their biological effects [73]. In this study, we used E171 containing about 40% nanoparticles (<100 nm) and 60% microparticles (>100 nm) [74], and results revealed strong organ injury. Previous work reported that both E171 components promote ROS and genotoxicity in human Caco-2 and HCT116 cells [75]. Thus, both nano- and microparticles may produce detrimental effects in organs from E171-exposed rats. On the other hand, ZnO NPs with a nanometric size of 50 nm were used. We previously showed E171 uptake by cardiac H9c2 cells [76]. There, ZnO NPs produced morphological alterations, decreased proliferation and viability, induced mitochondrial injury, and altered the redox state [77]. ZnO NPs release Zn^2+^ ions in acidic, aqueous, and biological environments, increasing their toxicity [78], predominantly by ROS generation [79]. Orally administered ZnO NPs deplete antioxidant molecules, promoting oxidative stress [80], causing damage to cellular structure and organelles [81,82]. Overall, ZnO NPs displayed higher toxicity than E171, possibly associated with their rapid absorption into circulation and biodistribution to multiple organs including the lung, liver, kidney, bones, brain, and spleen [83]. As we previously reported, ZnO NPs were absent in hearts derived from chronically exposed rats; however, there were strong morphological changes [84]. We hypothesized that free Zn^2+^ ions could be responsible. Cellular uptake of ZnO NPs occurs either as free Zn^2+^ ions, or as whole NPs [83]. ZnO NPs-exposed human bronchial epithelial cells (BEAS-2B) showed cellular NP accumulation followed by complete dissolution into intracellular Zn^2+^ ions complexed by molecular ligands [85]. Other studies support ZnO NPs release of toxic Zn^2+^ ions, entering cells through passive endocytosis [70,86]. Therefore, we hypothesize that oral exposure to ZnO NPs leads to absorption and bloodstream circulation, dissolution into Zn^2+^ ions, and cellular internalization causing organ damage.

In this work, NPs were dosed according to E171 levels found in humans. E171 intake is largely food-derived, and variations exist according to age and country. Dietary studies on E171 consumed in the USA and Great Britain indicate that children under 10 years old are more exposed than adults (1–3 vs. 0.2–1 mg of TiO_2_/kg of body weight (bw)/day, respectively), based on a higher consumption of E171-containing products including chewing gum and candy [87,88]. Daily E171 intake in Europe ranges from 0.2 to 5.5 mg/kg bw/day in people <11 months old and >65 years old [89]. In our experimental rats, 10 mg/kg bw/third day of E171 and ZnO NPs, equivalent to 5 mg/kg bw/day, was used [84].

Limitations: Our results reveal multi-organ injury caused by chronic oral exposure to metallic NPs. Further research is warranted on underlying molecular mechanisms causing organ and systemic damage. The present study was performed only in rats. Therefore, additional studies are required on alternative animal models and human post mortem samples. Despite these limitations, our results highlight that E171 and ZnO NPs consumption as food additives may represent a risk factor for chronic degenerative diseases.

## 4. Materials and Methods

### 4.1. E171 and ZnO NP Characterization

E171 was purchased from Mark Al Chemical de México (CAS number: 13463-67-7; color index 77891) (Mexico City, Mexico), and ZnO NPs (<50 nm) were obtained from Sigma Aldrich (677450) (St. Louis, MO, USA). E171 and ZnO NPs were suspended in an HEPES saline solution (150 mM NaCl, 10.9 mM HEPES, 4.4 mM KCl, 12.2 mM glucose; pH 7.4 before exposure). In previous work from our group [34], both NPs were suspended in three different media: the HEPES saline solution (pH 7.4), simulated gastric fluid (pH 1.1), and simulated intestinal fluid (pH 6.8). NPs were characterized by primary shape and physicochemical characteristics such as hydrodynamic size and zeta potential measured by dynamic light scattering with a Zetasizer nano-zs90. The primary size of NPs was determined by transmission electron microscopy (TEM) [34].

### 4.2. Rat Protocol

This research protocol on laboratory rats was approved by the Research Ethics Committee and Committee for Care and Use of Laboratory Animals of the Instituto Nacional de Cardiología Ignacio Chávez (protocol number 21-1274). Research was performed according to specifications of Norma Oficial Mexicana NOM-062-ZOO-1999.

Male Wistar rats from 2 months old, weighing approximately 200 g each, were used. Rats were divided into three groups of 6 rats each—group I: the saline solution (control), group II: E171, and group III: ZnO NPs. NPs were suspended in the HEPES saline solution at 1 mg/mL and kept at room temperature. Before use, NPs were vortexed at maximum speed for 5 min and then briefly between each rat to avoid their aggregation. The saline solution (control) and food additives were orally administered through an esophageal cannula at 10 mg/Kg every third day for three months. Rats were fed and hydrated ad libitum and weighed every week during exposure. After experiments, rats were sacrificed with sodium pentobarbital (120 mg/Kg) administered intraperitoneally. After euthanasia, organs were dissected and perfused with a phosphate-buffered solution (PBS, pH 7.4), to remove red blood cells and clots. Following excision, organs were fixed in 4% paraformaldehyde–PBS, gradually dehydrated in ethanol, cleaned in xylene, paraffin-embedded for HE staining and immunofluorescence, and fixed with 2% glutaraldehyde for the TEM analysis. All organs were kept at 4 °C.

### 4.3. Histology

All tissue samples were paraffin-embedded. Cross-sections from all organs (5 µm thick) were obtained from control and exposed (E171 and ZnO NPs) groups. The histological analysis was performed by hematoxylin–eosin (HE) staining.

### 4.4. TEM Analysis

Organ fragments were fixed with PBS containing 2.5% glutaraldehyde plus 2.5% paraformaldehyde, pH 7.4. Tissues were washed with PBS and post-fixed with osmium tetroxide (OsO4). Then, tissues were dehydrated at gradually increasing ethanol concentrations (50, 75, 85, 90, and 100%). Tissues were infiltrated with epoxy resin (Epon 812), then polymerized overnight at 60 °C. Subsequently, thin 240 nm sections were made and samples were selected; then, ultra-thin 60 nm sections were made and impregnated with uranyl acetate and lead acetate. Finally, samples were observed in a JEOL 10-10 transmission electron microscope.

### 4.5. Immunofluorescence

We cut 5 µm thick tissue sections and mounted them on coverslips for immunofluorescence staining. Sections were placed in slide holders containing a 250 mL retrieval buffer (0.5 mM EGTA in 10 mM Tris-base buffer, pH 9). Then, they were boiled in a microwave oven for 10 min. Sections were incubated with a 10% blocking serum in PBS for 1 h at room temperature followed by incubation at 4 °C overnight with a primary anti-ceramide antibody (C8104, Sigma), at a 1:500 dilution. After incubation, sections were washed and incubated with a secondary antibody, goat anti-mouse IgG-FITC, diluted at 1:200 at 4 °C for 4 h at room temperature (31569, Invitrogen, Waltham, MA, USA) [90]. Tissues were co-stained with nitrotyrosine (sc-32757 PE, Santa Cruz Biotechnology, Dallas, TX, USA) at a 1:500 dilution. Tissues were also co-stained with LAMP-2 (sc-71492, Santa Cruz Biotechnology) and subsequently incubated with the secondary antibody goat anti-mouse IgG H&L (Alexa Fluor^®^ 405, ab-175660, abcam, Cambridge, UK) diluted at 1:200 at 4 °C for 4 h at room temperature.

Negative controls were processed likewise but excluding the primary antibody. Sections were mounted on microscope coverslips using Vectashield. After the immunoassay, label detection for Cer, NT, and LAMP-2 was performed by acquiring images at 20x magnification in a FLoid, Cell Imaging Station (Life Technologies, Carlsbad, CA, USA). Four photographs of each tissue from 4 different rats were acquired; integrated optical density (IOD, lum/pix^2^) was quantified with Image-Pro-9 software from Media Cybernetics (Rockville, MD, USA). Graphs presented mean, standard deviation, and statistically significant values.

### 4.6. Statistical Analysis

Differences between exposed and control groups were measured by Student’s *t* test. Statistical significance was considered at *p* < 0.05.

## 5. Conclusions

Our results suggest a complex interplay between ROS, Cer, LAMP-2, and NT modulating organ functions during NP damage. Chronic exposure to E171 and ZnO NPs in rats produced histological and cellular alterations, up-regulating the expression of proteins associated with inflammation, stress, and organelle dysfunction, causing organ damage (NT, LAMP-2). This could eventually develop into organ failure and deadly diseases; therefore, the food-derived consumption of these NPs constitutes a hazard for human health.

## Figures and Tables

**Figure 1 ijms-25-05881-f001:**
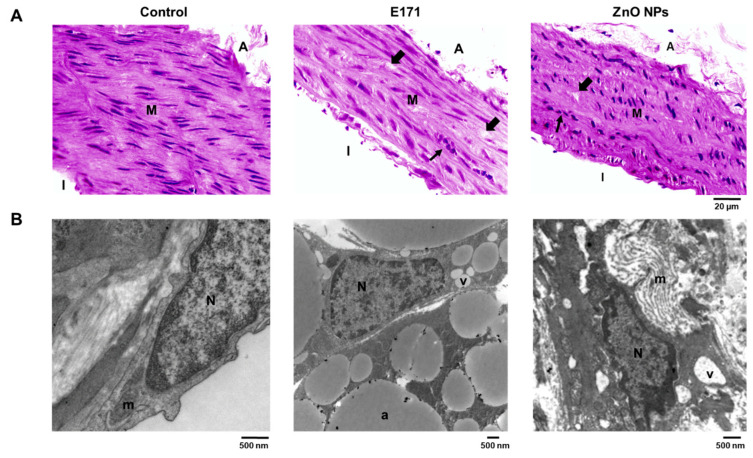
Dorsal aortic sections with HE staining (**A**) or analyzed by TEM (**B**) from controls and rats exposed to E171 and ZnO NPs. Microphotographs show representative results from one of six rats analyzed independently. A = adventitia, M = media, I = intima, v = vacuole, a = adipocytes, N = nucleus, m = mitochondria. Thick arrows = aneurysms, thin arrows = cellular infiltration.

**Figure 2 ijms-25-05881-f002:**
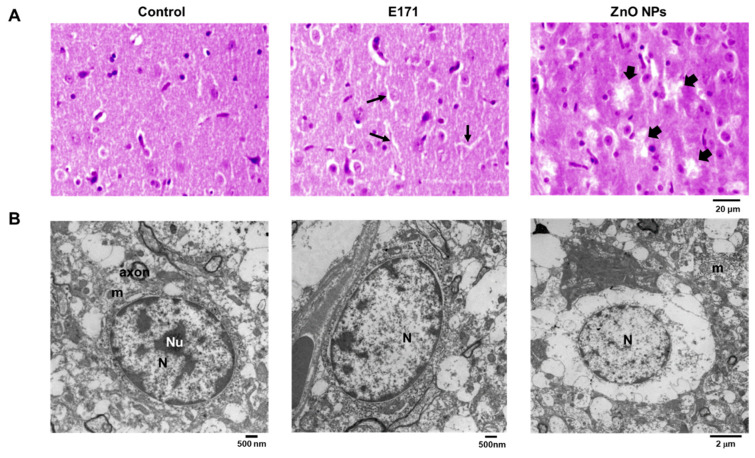
Coronal section of brain cortex with HE staining (**A**) or analyzed by TEM (**B**) from controls and rats exposed to E171 and ZnO NPs. Microphotographs show representative results from one of six rats analyzed independently. N = nucleus, Nu = nucleolus, m = mitochondria. Thin arrows = aneurysms in E171-exposed rats, thick arrows = aneurysms in ZnO NPs-exposed rats.

**Figure 3 ijms-25-05881-f003:**
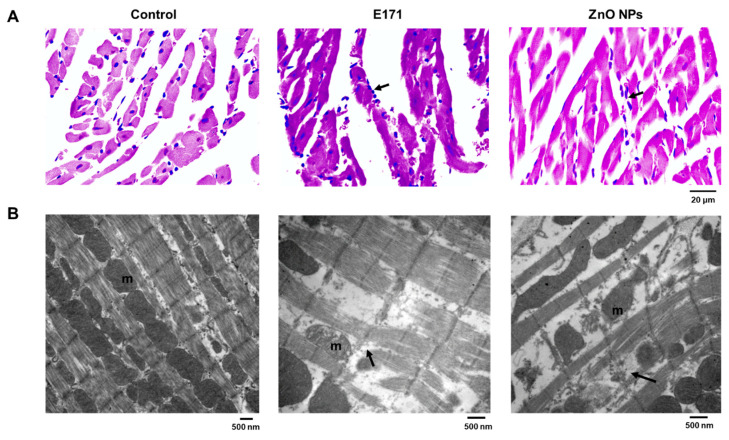
HE-stained cross-sections of heart ventricles (**A**) or analyzed by TEM (**B**) from control and exposed rats with E171 and ZnO NPs. Microphotographs show representative results from one of six rats analyzed independently. m = mitochondria.

**Figure 4 ijms-25-05881-f004:**
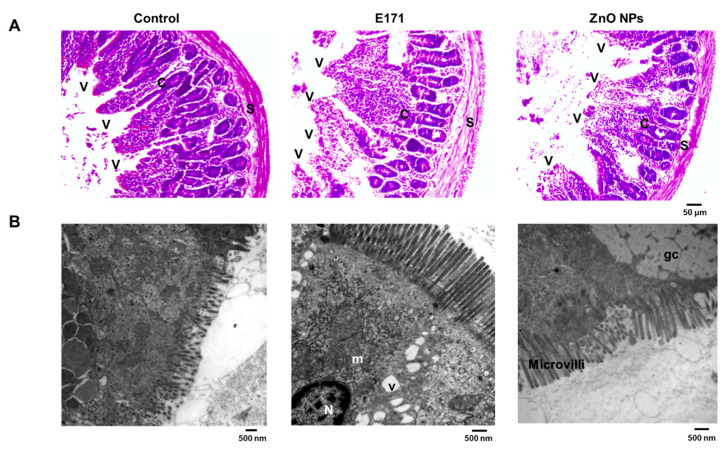
Small intestine cross-sections with HE staining (**A**) or analyzed by TEM (**B**) from controls and rats exposed to E171 and ZnO NPs. Microphotographs show representative results from one of six rats analyzed independently. V = villi, C = crypts, S = submucosa, gc = goblet cells, N = nucleus, m = mitochondria, and v = vacuole.

**Figure 5 ijms-25-05881-f005:**
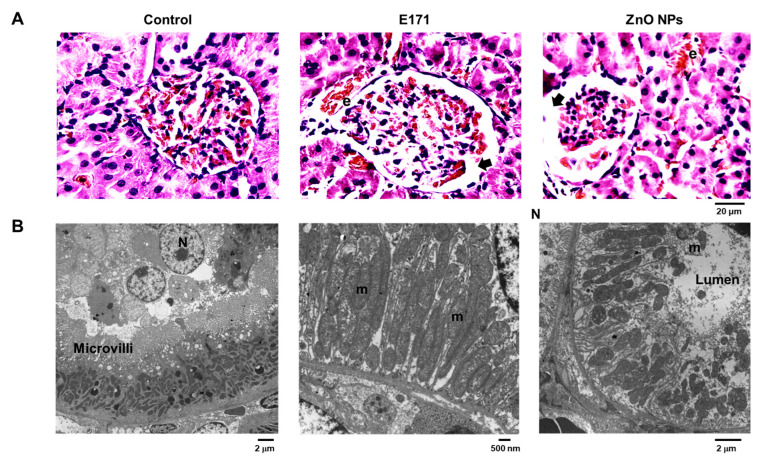
Kidney cross-sections with HE staining (**A**) or analyzed by TEM (**B**) from control and rats exposed to E171 and ZnO NPs. Microphotographs show representative results from one of six rats analyzed independently. Arrows indicate Bowman’s space and erythrocytes. N = nucleus, m = mitochondria, and e = erythrocytes.

**Figure 6 ijms-25-05881-f006:**
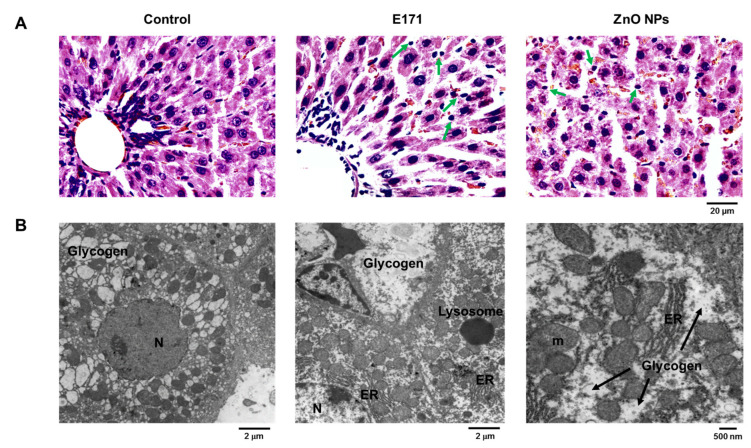
Liver cross-sections with HE staining (**A**) or analyzed by TEM (**B**) from control and rats exposed to E171 and ZnO NPs. Microphotographs show representative results from one of six rats analyzed independently. N = nucleus, m = mitochondria, and ER = endoplasmic reticulum. Green arrows indicate cellular infiltration.

**Figure 7 ijms-25-05881-f007:**
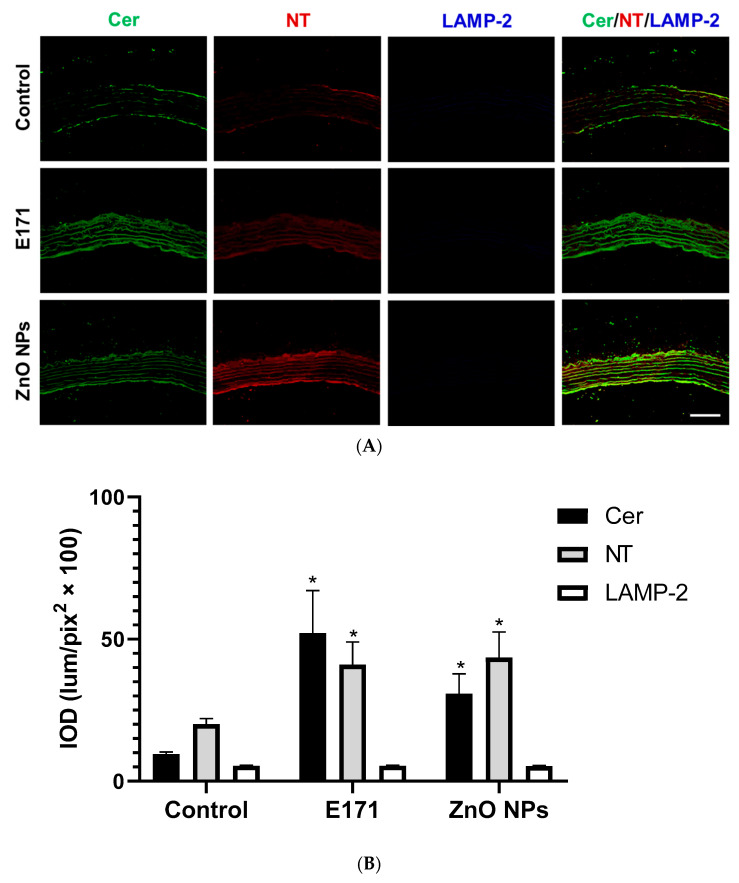
Immunofluorescence staining of dorsal aortic cells. (**A**) Representative images of ceramide (Cer, green), nitrotyrosine (NT, red), and LAMP-2 (blue) immunodetection in controls and rats exposed to E171 and ZnO NPs. Bar = 100 µm. (**B**) Integrated optical density (IOD) in lum/pix^2^ × 100. Data presented as mean ± standard deviation of 16 (4 areas × 4 rats) random areas of each group of rats. * *p* < 0.05 compared to control.

**Figure 8 ijms-25-05881-f008:**
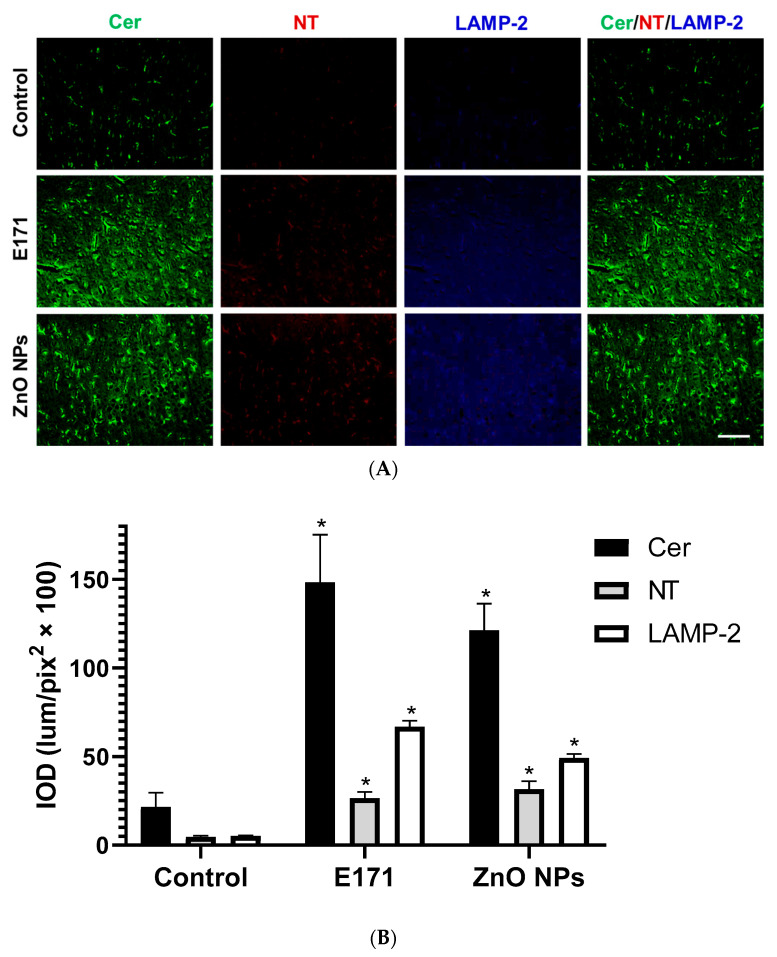
Immunofluorescence staining of brain cortex. (**A**) Representative images of ceramide (Cer, green), nitrotyrosine (NT, red), and LAMP-2 (blue) immunodetection in control and rats exposed to E171 and ZnO NPs. Bar = 100 µm. (**B**) Integrated optical density (IOD) in lum/pix^2^ × 100. Data presented as mean ± standard deviation of 16 (4 areas × 4 rats) random areas of each group of rats. * *p* < 0.05 compared to control.

**Figure 9 ijms-25-05881-f009:**
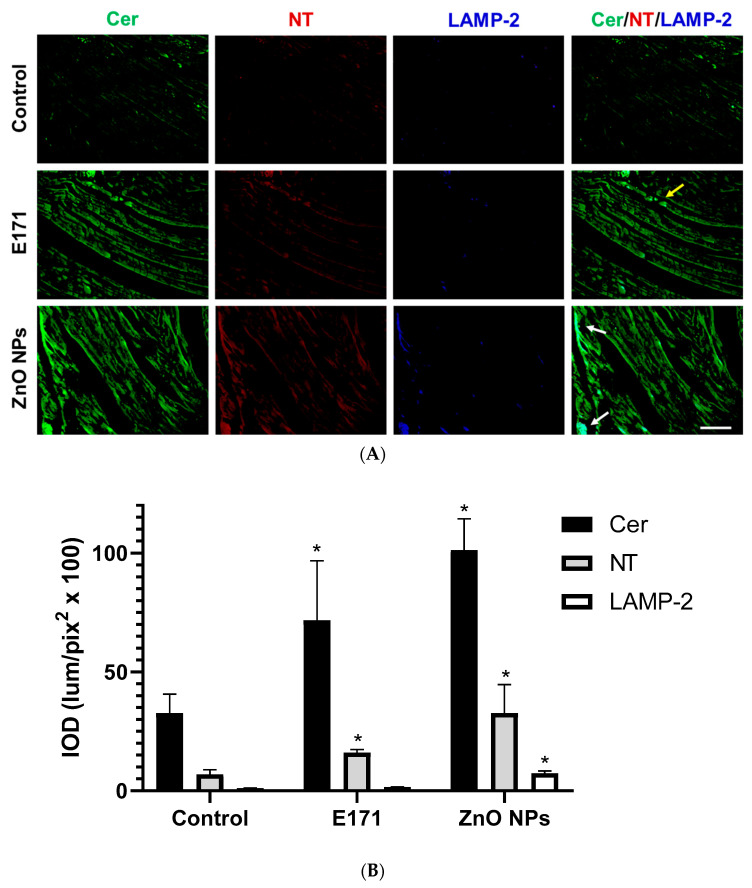
Immunofluorescence staining of myocardial fibers. (**A**) Representative images of ceramide (Cer, green), nitrotyrosine (NT, red), and LAMP-2 (blue) immunodetection in controls and rats exposed to E171 and ZnO NPs. Bar = 100 µm. (**B**) Integrated optical density (IOD) in lum/pix^2^ × 100. Data presented as mean ± standard deviation of 16 (4 areas × 4 rats) random areas of each group of rats. * *p* < 0.05 compared to control.

**Figure 10 ijms-25-05881-f010:**
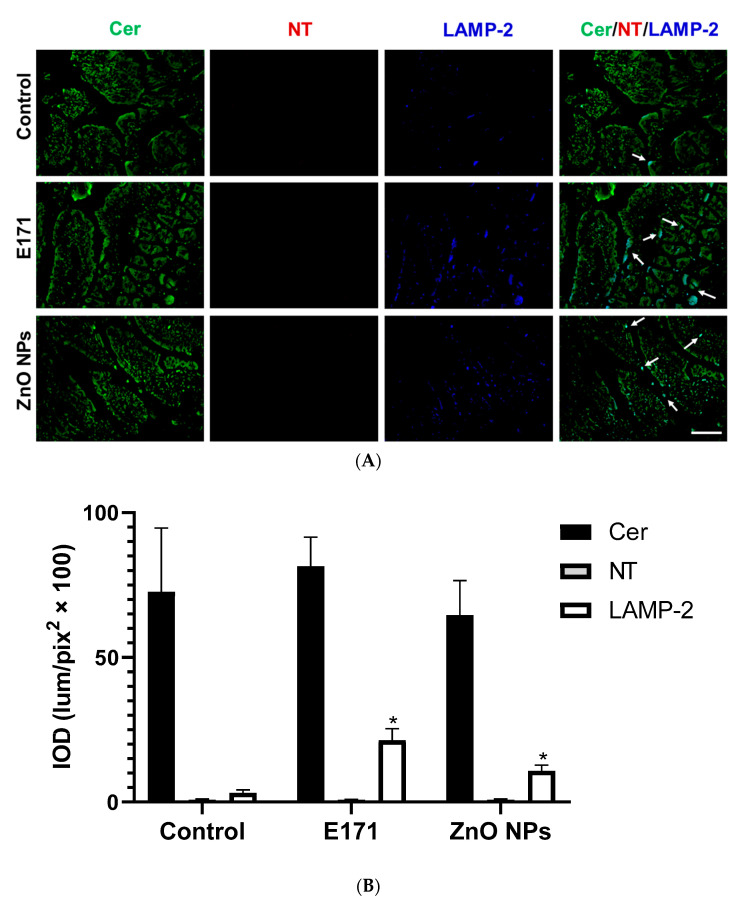
Immunofluorescence staining of small intestinal sections. (**A**) Representative images of ceramide (Cer, green), nitrotyrosine (NT, red), and LAMP-2 (blue) immunodetection in controls and rats exposed to E171 and ZnO NPs. Bar = 100 µm. (**B**) Integrated optical density (IOD) in lum/pix^2^ × 100. Data presented as mean ± standard deviation of 16 (4 areas × 4 rats) random areas of each group of rats. Arrows = LAMP-2 co-localized with Cer. * *p* < 0.05 compared to control.

**Figure 11 ijms-25-05881-f011:**
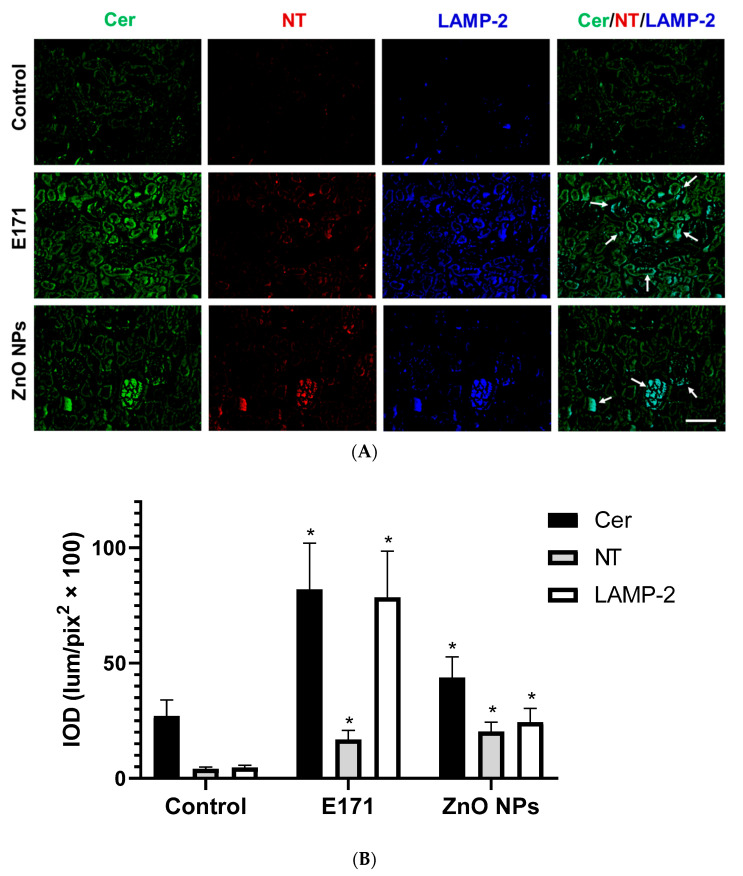
Immunofluorescence staining of kidney sections. (**A**) Representative images of ceramide (Cer, green), nitrotyrosine (NT, red), and LAMP-2 (blue) immunodetection in controls and rats exposed to E171 and ZnO NPs. Bar = 100 µm. (**B**) Integrated optical density (IOD) in lum/pix^2^ × 100. Data presented as mean ± standard deviation of 16 (4 areas × 4 rats) random areas of each group of rats. Arrows = NT, LAMP-2, and Cer co-localized. * *p* < 0.05 compared to control.

**Figure 12 ijms-25-05881-f012:**
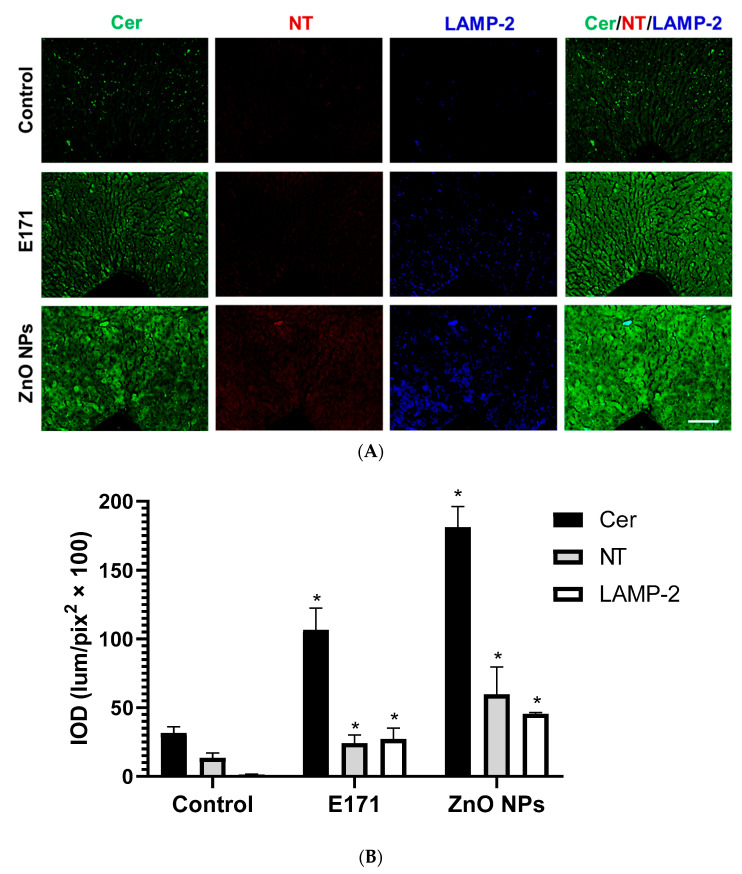
Immunofluorescence staining of liver sections. (**A**) Representative images of ceramide (Cer, green), nitrotyrosine (NT, red), and LAMP-2 (blue) immunodetection in controls and rats exposed to E171 and ZnO NPs. Bar = 100 µm. (**B**) Integrated optical density (IOD) in lum/pix^2^ × 100. Data presented as mean ± standard deviation of 16 (4 areas × 4 rats) random areas of each group of rats. * *p* < 0.05 compared to control.

## Data Availability

The authors confirm that data supporting findings of this study are available within the article and/or its Appendix A.

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
