# Peer review of "Oral Exposure to Titanium Dioxide E171 and Zinc Oxide Nanoparticles Induces Multi-Organ Damage in Rats: Role of Ceramide"

_ijms, 2024, doi:10.3390/ijms25115881_

Round 1
Reviewer 1 Report
Comments and Suggestions for Authors
Oral Exposure to Titanium Dioxide E171 and Zinc Oxide Nanoparticles Induces Multi-Organ Damage in Rats: Role of the Ceramide 4 Rocío Bautista-Pérez et al
Dear Editor-in-Chief (IJMS)
Comments to the Editor and authors
My Decision is minnor revision. However, before the final aceptation, these suggestions must be included in the final manuscript.
This study evaluates the toxicity of several organs from Wistar rats exposed orally for 90 days was analyzed by evaluating ceramide (Cer), nitrotyrosine (NT), and lysosome- associated membrane protein 2 (LAMP-2).
They osberved relevante morphological changes as disorganization, inflammatory cell infiltration and mitochondrial damage and also Cer, NT, and LAMP-2 were present in liver, kidney, and brain of ZnO NPs exposed rats, and in rat hearts under ZnO NPs exposure.
E171 up-regulated Cer and NT levels in aorta and heart, while ZnO NPs up-regulated these in aorta. Both NPs increased LAMP-2 expression in intestine.
This study demonstrate that chronic oral exposure to metallic NPs causes multiorgan injury.
-In general, shall you explain signaling pathways involved in detrimental effect of ceramide, in ZnO NPs up treated rats. These results are consistent although I miss these ZnNO nanoparticles altered signaling pathways.
In general, the design is correct but the discussion should include these observed anatomical differences among treatments. The metodology for the quantification of inmunofluorescence should be describe in more detail (number of analyzed sectons, intensity of fluorescence, etc). The magnification is not visible in these inmunofluorescences and figures 8 and 10 should indicate details with higher maganification (at least for relevance differences among markers and treatments).
The reference Merino JJ et al,. 2019 should be included in the reference section.
Merino JJ, Cabaña-Muñoz ME, Toledano Gasca A, Garcimartín A, Benedí J, Camacho-Alonso F, Parmigiani-Izquierdo JM. Elevated Systemic L-Kynurenine/L-Tryptophan Ratio and Increased IL-1 Betaand Chemokine (CX3CL1, MCP-1) Proinflammatory Mediators in Patients with Long-Term Titanium Dental Implants J Clin Med. 2019 Sep 2;8(9):1368. doi: 10.3390/jcm8091368)
methods
Since before use, NPs were vortexed at maximum speed for 5 min and then briefly between each rat to avoid their aggregation, shall you explain who thse NP were injected to ratswith more details. Thanks
Line 95. I was wondering if NP are not digested by acids in the stomach since it has been administered by a direct gastric canulation. In fact, the saline solution (control) and the two food additives were orally administered through an esophageal cannula at 10 mg/Kg every third day for three months. Does this long-term procedure provoke damage in tissues?
Line 93. Please, indicate the reason by wchih NPs were suspended in HEPES saline solution at 1 mg/mL and kept at room temperature
Line 98. Is there any possible interaction between anesthesia and NP here (agregation, precipitation...etc?
These inmunofluorescence are difficult to quantify, and paracine is not esay. It is a good election the use of 5 micrometer section. However, have used you a citrate buffer toenhance stainings with these secondary antibody? (goat anti-mouse IgG-FITC, m-IgGκ BP-PE, goat anti-mouse IgG H&L (Alexa Fluor® 405) diluted at 1:50 at 4°C for 4 h at room 125 temperature. In addition, Ig Gk is a rare secondary antibody. Shall you explain teh reason of your choice; Please, include a table with primary and secundary antibodies here (specie, fluorecrome, concentration, time of exposure, dilution, etc).
Describe the relevance of adipocytes in the figure that shows electron microscope
Line 163. This reviewer does not understant the meaning of holes in this sentence… However, brains from E171-treated rats (thin arrows) and from those exposed to ZnO NPs (thick arrows). Are ¨these holes¨ a problema during the preparation of parafine sections in your study?
Lin1 168. The descripton of cristals is not clear for me. They indicate ¨ On the other hand, brains from ZnO NPs-exposed rats, exhibited mitochondrial injured as lower numbers of cristae indicating mitochondrial dysfunction and brain damage
Explain how NP alters the Bowman capsule in the kidney and also precise the anatonical part analyzed in kidneys (tubular cells, Bowman capsule, .etc).
Please, indicate how inmunofluorescce for ceramide has been quantified in your study. Also explain possible downstream signaling pathways involved in these ceramide changes in your study or possible suggestions if exist evidences.
The inmunofluorescences from figure-8 and 12 must include a high magnificatiion for ceramide/NT/or LAM-2 (at least for representative differences among treatments). How many slices were analyzed here? Please, also put the magnifications (is not visible) in all figures (10x or 4X).
Following the same criteria, indicate the exactly anatomical area evaluated for kidney analysis in this figure 10 (tubular cells or Bowman capsule)? Also include a table for all characteristic of primary and secondary antibodies used for inmunofluorescence study in all evaluated tissues.
Also, explain morphology changes found in the aorta and posible pathological mecahsnism by E17 or nanoparticles.
The discussion must describe these alterations among treatments in different tissues by Nanoparticles or E171. For example, the desorganization of glomerulus from kidney could led to kidney dysfuction. The cortex alterations could accelerate the debut of neurodegenerative or neuropsychiatric disease. Please, discuss these evidences with your treatmentes in more detail by including published findings with nanoparticies. From this perspective, introduction msut containt the ¨old¨ inert concept of Zn or Titanium nanoparticles as inert biomaterials without toxicity. However, the ¨new evidences¨ suggesting a detrimental role of these high Zn levels or titanium biomateriales should be also included here; please, add this cyte Merino et al 2019 after lines 325-330, which support that CX3CL1=fractalkine or MCP-1 contribute to generate a silent and chronic systemic iinflammatory state in patients with long-term titanium dental implants (Merino et al.,2019). Thes evidence support the connexion between titanium and detrimental effects in patients (Merino JJ, Cabaña-Muñoz ME, Toledano Gasca A, Garcimartín A, Benedí J, Camacho-Alonso F, Parmigiani-Izquierdo JM. Elevated Systemic L-Kynurenine/L-Tryptophan Ratio and Increased IL-1 Betaand Chemokine (CX3CL1, MCP-1) Proinflammatory Mediators in Patients with Long-Term Titanium Dental Implants J Clin Med. 2019 Sep 2;8(9):1368. doi: 10.3390/jcm8091368). Include within the reference section.
Line 345. I agree with the described role of ceramide although I miss signalling pathways involved in these anatomical alterations by high levels of ceramide. Include evidences that support your findings with ceramide and LAMP.
In my opinion, the suggested mechanism of ROS production and nitrosamines in the aortha seems to be a general mechanism for all tissues analyzed here. Is this true except for the aorta? Please, shall you explain why LAM-2 is not upregulated in the aorta
Line 376. They also indicate that ¨The present study analyzed effects of food-grade titanium dioxide (E171), containing about 40% nanoparticles (100 nm) [63] revealing strong organ injury. Did you quantifie4d these different percenttages in the treated mices in your study?
Line 388.Also, explain the ZnO NPs in hearts derived from chronically exposed rats altgough there is morphological alterations (paper Cardiovascular Toxicology 2). Is this another section of a spetial issue?
Are E171 limit level identical for all countries in the world?.
Line 404. They also indicate ¨In experimental rats, 10 mg/kg bw/third day of E171 and ZnO NPs, equivalent to 5 mg/kg bw/day¨. Shall you include the reference.
The discussion must explain these differential anatomical findings observed among E171 and Zn-nanoparticles treated mices, inclugins some signalling pathways (in general).
Include this cyte Merino et al., 2019 in the reference section. Thanks¡
Finally, include the reference in the bibliography section by including Merino JJ, Cabaña-Muñoz ME, Toledano Gasca A, Garcimartín A, Benedí J, Camacho-Alonso F, Parmigiani-Izquierdo JM. Elevated Systemic L-Kynurenine/L-Tryptophan Ratio and Increased IL-1 Betaand Chemokine (CX3CL1, MCP-1) Proinflammatory Mediators in Patients with Long-Term Titanium Dental Implants J Clin Med. 2019 Sep 2;8(9):1368. doi: 10.3390/jcm8091368)
Comments on the Quality of English Language
The english style should be improved.
Author Response
May 13th, 2024.
Dear reviewer:
I thank you to all your comments which have enriched and improved our work. We hope to satisfactorily answer all the concerns. We send you our revised manuscript (marked in yellow), and the response to your comments detailed point-by-point.
Reviewer 1:
Oral Exposure to Titanium Dioxide E171 and Zinc Oxide Nanoparticles Induces Multi-Organ Damage in Rats: Role of the Ceramide 4 Rocío Bautista-Pérez et al
Dear Editor-in-Chief (IJMS)
Comments to the Editor and authors
My Decision is minnor revision. However, before the final aceptation, these suggestions must be included in the final manuscript.
This study evaluates the toxicity of several organs from Wistar rats exposed orally for 90 days was analyzed by evaluating ceramide (Cer), nitrotyrosine (NT), and lysosome- associated membrane protein 2 (LAMP-2).
They osberved relevante morphological changes as disorganization, inflammatory cell infiltration and mitochondrial damage and also Cer, NT, and LAMP-2 were present in liver, kidney, and brain of ZnO NPs exposed rats, and in rat hearts under ZnO NPs exposure.
E171 up-regulated Cer and NT levels in aorta and heart, while ZnO NPs up-regulated these in aorta. Both NPs increased LAMP-2 expression in intestine.
This study demonstrate that chronic oral exposure to metallic NPs causes multiorgan injury.
-In general, shall you explain signaling pathways involved in detrimental effect of ceramide, in ZnO NPs up treated rats. These results are consistent although I miss these ZnNO nanoparticles altered signaling pathways.
Answer: All the possible signaling pathways of ceramide were added in the Discussion section.
In general, the design is correct but the discussion should include these observed anatomical differences among treatments. The metodology for the quantification of inmunofluorescence should be describe in more detail (number of analyzed sectons, intensity of fluorescence, etc). The magnification is not visible in these inmunofluorescences and figures 8 and 10 should indicate details with higher maganification (at least for relevance differences among markers and treatments).
Answer: The anatomical differences were added to the Discussion section, which was completely re-structured. Quantification of immunodetection intensity has been added to Materials and Methods section (point 4.5).
The images attached in the complementary material show close-ups in which the details of all the immunofluorescence are distinguished.
The reference Merino JJ et al,. 2019 should be included in the reference section.
Answer: We thank your suggestion to add this reference because it is a very important founding, however, we think that is not suitable to our work because Merino and collaborators evaluated the effect of long-term titanium implants (the metal alone plus amalgams) on several systemic cytokines, chemokines and soluble fractalkine, but they did not use titanium dioxide nanoparticles, which are very different, both in their composition and in nanometer size. Therefore, their biological effects are not comparable.
Merino JJ, Cabaña-Muñoz ME, Toledano Gasca A, Garcimartín A, Benedí J, Camacho-Alonso F, Parmigiani-Izquierdo JM. Elevated Systemic L-Kynurenine/L-Tryptophan Ratio and Increased IL-1 Betaand Chemokine (CX3CL1, MCP-1) Proinflammatory Mediators in Patients with Long-Term Titanium Dental Implants J Clin Med. 2019 Sep 2;8(9):1368. doi: 10.3390/jcm8091368)
Methods:
Since before use, NPs were vortexed at maximum speed for 5 min and then briefly between each rat to avoid their aggregation, shall you explain who thse NP were injected to rats with more details. Thanks
Answer: NPs were administered to rats orally with an esophageal cannula and the whole description is included in Materials and Methods section (point 4.2). NPs were not injected.
Line 95. I was wondering if NP are not digested by acids in the stomach since it has been administered by a direct gastric canulation. In fact, the saline solution (control) and the two food additives were orally administered through an esophageal cannula at 10 mg/Kg every third day for three months. Does this long-term procedure provoke damage in tissues?
Answer: In this study we used a control group which was administered just saline solution in a similar way to the other groups. All results showed important changes in the tissue morphology and in the expression of all evaluated molecules in exposed rats compared with the control group, which indicates that exposition of rats to NPs caused organ damage.
Line 93. Please, indicate the reason by wchih NPs were suspended in HEPES saline solution at 1 mg/mL and kept at room temperature
Answer: In previous works of our work group, we have observed that the best solution to suspend NPs is phosphates buffer solution. The addition of Hepes to this solution avoids its changes of pH, achieving a better buffer solution and getting better its stability. NPs suspended in Hepes saline solution are stable to room temperature.
Line 98. Is there any possible interaction between anesthesia and NP here (agregation, precipitation...etc?
Answer: An interaction between pentobarbital and NPs has not been described. Furthermore, the anesthesia was used until the end of the experiment, therefore, we believe that it is not important to the obtained results.
These inmunofluorescence are difficult to quantify, and paracine is not esay. It is a good election the use of 5 micrometer section. However, have used you a citrate buffer toenhance stainings with these secondary antibody? (goat anti-mouse IgG-FITC, m-IgGκ BP-PE, goat anti-mouse IgG H&L (Alexa Fluor® 405) diluted at 1:50 at 4°C for 4 h at room 125 temperature. In addition, Ig Gk is a rare secondary antibody. Shall you explain teh reason of your choice; Please, include a table with primary and secundary antibodies here (specie, fluorecrome, concentration, time of exposure, dilution, etc).
Answer: A more detailed description of immunofluorescence assay and all characteristics of the antibodies were included in the Materials and Methods section, point 4.5.
Describe the relevance of adipocytes in the figure that shows electron microscope
Answer: One of the parameters that help us understand proper cellular functioning are adipocytes. Currently, adipocytes are considered a secretory organ with metabolic, endocrine, and regulatory functions. They synthesize molecules such as leptin and adiponectin (which have endocrine and inflammatory functions and play an important role in insulin resistance). In addition, adipose tissue secretes cytokines, mainly inflammatory such as tumor necrosis factor alpha (TNF-α) and interleukins 6 (IL-6), which is why the adipocyte plays an important role in chronic degenerative diseases. The imbalance in the production of adipokines is linked to the existence of metabolic syndrome and cardiovascular diseases. In this way, the volumes of fat vacuoles can increase or decrease depending on their functional state.
Line 163. This reviewer does not understant the meaning of holes in this sentence… However, brains from E171-treated rats (thin arrows) and from those exposed to ZnO NPs (thick arrows). Are ¨these holes¨ a problema during the preparation of parafine sections in your study?
Answer: The presence of spaces observed with HE staining and indicated in Figure 2A in the cerebral cortex of animals treated with E171 and ZnO, correspond to aneurysms (Dai et al., 2005). This cannot be attributed to a problem with the preparation of the samples for paraffin sections, since the tissue from control animals was processed in parallel and do not present this characteristic.
Lin1 168. The descripton of cristals is not clear for me. They indicate ¨ On the other hand, brains from ZnO NPs-exposed rats, exhibited mitochondrial injured as lower numbers of cristae indicating mitochondrial dysfunction and brain damage
Answer: In these micrographs with magnification of 20,000 and 50,000, mitochondrial damage and a low number of its cristae are clearly observed. Which implies a decrease and alteration of the functions of neuronal metabolism, in which a much greater energy expenditure is required than that required in other tissues, thus altering the electron and ATP transport chain.
Explain how NP alters the Bowman capsule in the kidney and also precise the anatonical part analyzed in kidneys (tubular cells, Bowman capsule, .etc).
Answer: A more detailed description about the effect of NPs affect the Bowman capsule in the kidney was added in the Discussion section.
Please, indicate how inmunofluorescce for ceramide has been quantified in your study. Also explain possible downstream signaling pathways involved in these ceramide changes in your study or possible suggestions if exist evidences.
Answer: A more detailed description of immunofluorescence assay for ceramide was included in the Materials and Methods section, point 4.5. The possible signaling pathways involved with changes in ceramide levels induced by nanoparticles were included in the Discussion section.
The inmunofluorescences from figure-8 and 12 must include a high magnificatiion for ceramide/NT/or LAM-2 (at least for representative differences among treatments). How many slices were analyzed here? Please, also put the magnifications (is not visible) in all figures (10x or 4X).
Answer: Images with a higher resolution were added as Supplementary Material in order to observe better the obtained results.
Four fields of 4 tissues from different animals were quantified (details are included in the methods section).
For immunofluorescence, all images were acquired at 20X and the corresponding scale is indicated in each figure legend [(Bar =100µm) or (Bar=50µm) in the supplementary material section].
Following the same criteria, indicate the exactly anatomical area evaluated for kidney analysis in this figure 10 (tubular cells or Bowman capsule)? Also include a table for all characteristic of primary and secondary antibodies used for inmunofluorescence study in all evaluated tissues.
Answer: The anatomical area evaluated for kidney analysis was indicated in the figure legend. All characteristics of the primary and secondary antibodies used were included in the Materials and Methods section, point 4.5.
Also, explain morphology changes found in the aorta and posible pathological mecahsnism by E17 or nanoparticles.
Answer: The morphological changes observed in the aorta from exposed rats to NPs were analyzed with more detail in the Discussion section.
The discussion must describe these alterations among treatments in different tissues by Nanoparticles or E171. For example, the desorganization of glomerulus from kidney could led to kidney dysfuction. The cortex alterations could accelerate the debut of neurodegenerative or neuropsychiatric disease. Please, discuss these evidences with your treatmentes in more detail by including published findings with nanoparticies. From this perspective, introduction msut containt the ¨old¨ inert concept of Zn or Titanium nanoparticles as inert biomaterials without toxicity. However, the ¨new evidences¨ suggesting a detrimental role of these high Zn levels or titanium biomateriales should be also included here; please, add this cyte Merino et al 2019 after lines 325-330, which support that CX3CL1=fractalkine or MCP-1 contribute to generate a silent and chronic systemic iinflammatory state in patients with long-term titanium dental implants (Merino et al.,2019). Thes evidence support the connexion between titanium and detrimental effects in patients (Merino JJ, Cabaña-Muñoz ME, Toledano Gasca A, Garcimartín A, Benedí J, Camacho-Alonso F, Parmigiani-Izquierdo JM. Elevated Systemic L-Kynurenine/L-Tryptophan Ratio and Increased IL-1 Betaand Chemokine (CX3CL1, MCP-1) Proinflammatory Mediators in Patients with Long-Term Titanium Dental Implants J Clin Med. 2019 Sep 2;8(9):1368. doi: 10.3390/jcm8091368). Include within the reference section.
Answer: The alterations observed in different tissues such as aorta, intestine, brain, liver were explained with more detail in the Discussion section.
A sentence of the inert effect of these materials was added in the Introduction section (at the beginning of the first paragraph).
We thank your suggestion to add this reference, but we think that is not suitable because Merino and collaborators evaluated the effect of long-term titanium implants (the metal alone plus amalgams) on several systemic cytokines, chemokines and soluble fractalkine, but they did not use titanium dioxide nanoparticles, which are very different, both in their composition and in nanometer size. Therefore, their biological effects are not comparable.
Line 345. I agree with the described role of ceramide although I miss signalling pathways involved in these anatomical alterations by high levels of ceramide. Include evidences that support your findings with ceramide and LAMP.
Answer: The possible role of ceramide, NT and LAMP-2 in the organ alterations induced by nanoparticles was added in the Discussion section.
In my opinion, the suggested mechanism of ROS production and nitrosamines in the aortha seems to be a general mechanism for all tissues analyzed here. Is this true except for the aorta? Please, shall you explain why LAM-2 is not upregulated in the aorta
Answer: This is a good question; However, we unknown why LAMP-2 expression was not detected in the aorta. It will be very interesting to explore deeply what happens in aortas from exposed rats to nanoparticles in next research.
Line 376. They also indicate that ¨The present study analyzed effects of food-grade titanium dioxide (E171), containing about 40% nanoparticles (100 nm) [63] revealing strong organ injury. Did you quantifie4d these different percenttages in the treated mices in your study?
Answer: The size distribution of E171 used in this work was previously determined by collaborators and is: 50-100 nm = 38%; 100-150 nm = 34%; 150-200 nm: 16%; and 200-250 nm = 12%. In other words, E171 has a 38% of NPs and 62% of MPs displaying an amorphous morphology (Mutagenesis, 2016, 00, 1–11; doi:10.1093/mutage/gew051). The size distribution of these nanoparticles does not change after their administration; therefore, this was not quantified in the rats.
Line 388.Also, explain the ZnO NPs in hearts derived from chronically exposed rats altgough there is morphological alterations (paper Cardiovascular Toxicology 2). Is this another section of a spetial issue?
Answer: These results were obtained by one previous work of our work group. This reference was added in the Discussion section (marked in yellow, reference 68).
Are E171 limit level identical for all countries in the world?.
Answer: No, the allowed limit to the use of E171 can vary. There are countries where its use is not allowed and others where it is still used as México; therefore, we considered important to evaluate its effect in an in vivo model.
Line 404. They also indicate ¨In experimental rats, 10 mg/kg bw/third day of E171 and ZnO NPs, equivalent to 5 mg/kg bw/day¨. Shall you include the reference.
Answer: This same dose was administrated in our rat model previously by our work group. This reference was included in the text.
The discussion must explain these differential anatomical findings observed among E171 and Zn-nanoparticles treated mices, inclugins some signalling pathways (in general).
Answer: These differences between E171 and ZnO NPs were included in the Discussion section.
Include this cyte Merino et al., 2019 in the reference section. Thanks¡
Answer: We thank your suggestion to add this reference, but we think that is not suitable for our work, we already explained it in another point before.
Finally, include the reference in the bibliography section by including Merino JJ, Cabaña-Muñoz ME, Toledano Gasca A, Garcimartín A, Benedí J, Camacho-Alonso F, Parmigiani-Izquierdo JM. Elevated Systemic L-Kynurenine/L-Tryptophan Ratio and Increased IL-1 Betaand Chemokine (CX3CL1, MCP-1) Proinflammatory Mediators in Patients with Long-Term Titanium Dental Implants J Clin Med. 2019 Sep 2;8(9):1368. doi: 10.3390/jcm8091368)
Answer: We thank your suggestion to add this reference, but we think that is not suitable for our work, we already explained it in another point before.
Sincerely,
Dr. Rebeca López-Marure
Corresponding author

Reviewer 2 Report
Comments and Suggestions for Authors
The current manuscript from Dr. López-Marure’s research group explored the health impact of titanium dioxide (E171) and zinc oxide nanoparticles (ZnO NPs) using Wistar rats as the model.
The study design is very superficial, and more elaborate analyses and a comprehensive experimental design are required to confirm the results and understand the mechanistic processes involved.
Suggestions
1. Rats were exposed to the treatments using an esophageal cannula. To properly investigate the research question, authors must figure out a way to determine the distribution and systemic availability of the NPs. Is there a way to determine the presence of the NPs in the blood?
2. Please provide the rationale of the dosage used in the study. Authors used a dose of 10 mg/Kg of animal body weight. Did authors perform a preliminary study determining the dose-response. What is the rationale behind the use of this dose?
3. It is concerning that similar morphological features in aorta and brain are described differently by the authors. The presence of empty spaces in the aorta has been termed aneurysms, yet in brain, similar structures are termed as “holes”. Is it possible that the presence of NPs somehow affected the tissue preservation, fixation, and subsequent processing of the histological sections?
4. If possible, authors are requested to provide some form of quantification of tissue damage parameter or indices, using histological images. Currently, the observations are mostly derived from the representative images.
5. The conclusions associated with the impact of NPs on the glycogen deposition in the liver should be revisited. Authors are requested to perform a secondary measurement of glycogen content in the liver. The current description of the glycogen deposition in the liver is not convincing.
6. It is very hard to see the LAMP2 immunofluorescence in the images. In most cases, no expression (blue immunofluorescence) is visible in any of the exposure groups. Hence a conclusion of “no change” cannot be ascertained. Authors must address this aspect as these results are very crucial to explore the study objectives.
7. Since only male rats were used, an obvious question arises about the effect of gender in the observed impacts.
8. Methods, 2.5. Immunohistochemistry, please provide more information on the assessment of the immunofluorescence to compare across the exposure groups.
9. Figure 8A, the merged field (Cer/NT/LAMP-2) for control group is not convincing. The individual images are not adding up to form the merged field.
10. Did authors notice any change in the animal body weight during the exposure duration?
11. Please expand the Methods description of the Statistical Analysis used in the study to analyze the results.
12. A “limitations” section in the discussion will be very helpful.
Other comments
1. Please define the abbreviations at their first mention. For example, please define “NP” as nanoparticles before using the abbreviation in the manuscript.
2. Authors are requested to edit the abstract text to clearly indicate the study design. Since they have used three groups, including two NP exposure groups, the description of the results should reflect this.
3. Current study design only uses 90 days’ time point. A few more intermediate time points would be very supportive.
4. Please report the catalogue numbers of all the reagents used in the study. Especially, the antibodies used in histological examinations.
5. Not sure what authors meant by “Nanotechnology is mattering manipulation at nanometric scales”.
6. Please check that all open parentheses are closed.
7. The manuscript will benefit greatly if edited for language use.
Comments on the Quality of English LanguageModerate editing needed.
Author Response
May 13th, 2024.
Dear reviewer:
I thank you to all your comments which have enriched and improved our work. We hope to satisfactorily answer all the concerns. We send you our revised manuscript (marked in yellow), and the response to your comments detailed point-by-point.
Reviewer 2:
The current manuscript from Dr. López-Marure’s research group explored the health impact of titanium dioxide (E171) and zinc oxide nanoparticles (ZnO NPs) using Wistar rats as the model.
The study design is very superficial, and more elaborate analyses and a comprehensive experimental design are required to confirm the results and understand the mechanistic processes involved.
Suggestions
- Rats were exposed to the treatments using an esophageal cannula. To properly investigate the research question, authors must figure out a way to determine the distribution and systemic availability of the NPs. Is there a way to determine the presence of the NPs in the blood?
Answer: As you mentioned, this work is very general. Our main objective was analyzing the histological and cell changes and to determine the alteration of several molecules involved with damage in various organs derived from chronically exposed rats to metal nanoparticles. The distribution and systemic availability of these NPs as well as their presence in the blood have already determined in other works (cited references in the introduction section). However, our results demonstrate for first time important tissue alterations and up-regulation in the expression of molecules related with damage, indicating that the oral consumption of these NPs can induce organic dysfunction.
- Please provide the rationale of the dosage used in the study. Authors used a dose of 10 mg/Kg of animal body weight. Did authors perform a preliminary study determining the dose-response. What is the rationale behind the use of this dose?
Answer: The used dose was determined according to the concentrations found of these nanomaterials in human during their consumption (around 5 mg/kg/day). This is indicated in the last paragraph of the Discussion section. We previously performed another study using the same dose of this work were observed important cardiac alterations (reference 34). The idea of using this dose was to simulate what a human consumes daily.
- It is concerning that similar morphological features in aorta and brain are described differently by the authors. The presence of empty spaces in the aorta has been termed aneurysms, yet in brain, similar structures are termed as “holes”. Is it possible that the presence of NPs somehow affected the tissue preservation, fixation, and subsequent processing of the histological sections?
Answer: Thanks for the observation, the description has been unified, referring in both cases (aorta and brain) as aneurysms (Dai et al., 2005). We do not know if the NPs as such reach the brain and we cannot rule out that the E171 and ZnO NPs and/or their derivatives enter the circulation changing the cyto-architecture of different tissues that in the aorta and cerebral vasculature are reflected as aneurysms. We rule out that this is a direct effect on the processing of the samples, rather it reflects alteration in the tissue of the animals. treated with NPs compared to control tissues.
- If possible, authors are requested to provide some form of quantification of tissue damage parameter or indices, using histological images. Currently, the observations are mostly derived from the representative images.
Answer: In the present study, the quantification of the immunodetection of the expression of ceramide (membranes), nitrotyrosine (nitrosating stress) and LAMP-2 (lysosome-associated membrane proteins) was considered as a reference for tissue damage related to membrane turnover, such as possible response associated with the endomembranous system in the presence of NPs
- The conclusions associated with the impact of NPs on the glycogen deposition in the liver should be revisited. Authors are requested to perform a secondary measurement of glycogen content in the liver. The current description of the glycogen deposition in the liver is not convincing.
Answer: A deeper discussion about the alteration of glycogen deposits found in livers from NPs exposed rats was added in the Discussion section. We consider that it will be very interesting to evaluated biochemistry the glycogen deposits in the liver, as well as other biochemical markers of damage in all organs; however, results obtained in this work are just a very general observation of what happens to the organs in rats chronically exposed orally to these nanoparticles.
- It is very hard to see the LAMP2 immunofluorescence in the images. In most cases, no expression (blue immunofluorescence) is visible in any of the exposure groups. Hence a conclusion of “no change” cannot be ascertained. Authors must address this aspect as these results are very crucial to explore the study objectives.
Answer: Tissues exposed showed signal compared with no exposed. To observe better the changes in the signal, the bright was adjusted in the images from tissues where the signal is weak (brain, heart, intestine) in the same proportion to control and treated.
- Since only male rats were used, an obvious question arises about the effect of gender in the observed impacts.
Answer: Investigation of potential sex-differences in the adverse health outcomes associated with nanoparticle is greatly lacking. Only ~20% of basic research in the general sciences use both male and female animals and a substantial percentage of these do not address differences between sexes within their analyses. However, to avoid possible sex-differences in hormone signaling, we always used male rats. It will be very interesting to use female rats in comparison to improve our understanding of sex as a biological variable in nanoparticles exposure.
- Methods, 2.5. Immunohistochemistry, please provide more information on the assessment of the immunofluorescence to compare across the exposure groups.
Answer: Nitrotyrosine (red) and LAMP-2 (blue) was performed by acquiring images at 20X magnification in an FLoid, Cell Imaging Station (Life Technologies, Carlsbad, CA, USA). Four photographs of each tissue from 4 different animals were acquired; on which the integrated optical density (IOD, lum/pix2) was quantified with the Image-Pro-9 software from Media Cybernetics. The graphs show the average values with the standard deviation and the corresponding significance values.
- Figure 8A, the merged field (Cer/NT/LAMP-2) for control group is not convincing. The individual images are not adding up to form the merged field.
Answer: Thanks for the observation, the image has been corrected.
- Did authors notice any change in the animal body weight during the exposure duration?
Answer: The animal body weight was determined in another experiment performed by our work group, where a reduction more than 10% was observed after 5 weeks of E171 exposure until end of treatment. ZnO NPs just reduced body weight after five and seven weeks, recovering after nine weeks (Correa Segura F. et al., Nanotoxicology 2024, 18:2, 122).
- Please expand the Methods description of the Statistical Analysis used in the study to analyze the results.
Answer: The description of the statistical analysis used was expanded in Material and Methods section (point 4.6).
- A “limitations” section in the discussion will be very helpful.
Answer: A limitations section was added in the last paragraph of the discussion.
Other comments
- Please define the abbreviations at their first mention. For example, please define “NP” as nanoparticles before using the abbreviation in the manuscript.
Answer: The abbreviation to nanoparticles (NPs) was already added in the first sentence of the Introduction section.
- Authors are requested to edit the abstract text to clearly indicate the study design. Since they have used three groups, including two NP exposure groups, the description of the results should reflect this.
Answer: This information was added in the Abstract.
- Current study design only uses 90 days’ time point. A few more intermediate time points would be very supportive.
Answer: The objective of this study was to evaluate the effect of these nanoparticles after an oral chronic exposure, to simulate the exposure that humans generally have to these nanoparticles. It will be very interesting to determine their effect after acute and short times of exposure.
- Please report the catalogue numbers of all the reagents used in the study. Especially, the antibodies used in histological examinations.
Answer: All information about antibodies used was added in the Materials and Methods section, point 4.5.
- Not sure what authors meant by “Nanotechnology is mattering manipulation at nanometric scales”.
Answer: We used this sentence because Nanotechnology definition is the branch of science and engineering devoted to designing, producing, and using structures, devices, and systems by manipulating atoms and molecules at nanoscale, i.e. having one or more dimensions of the order of 100 nanometres (100 millionth of a millimetre) or less.
- Please check that all open parentheses are closed.
Answer: This point was checked.
- The manuscript will benefit greatly if edited for language use.
Answer: The language of the whole text was checked carefully again.
Sincerely,
Dr. Rebeca López-Marure
Corresponding author

Round 2
Reviewer 2 Report
Comments and Suggestions for Authors
Please include a separate "limitations" section in the manuscript.
Comments on the Quality of English Languagestill needs improvement
Author Response
Dear reviewer:
I thank you to all your comments which have enriched and improved our work. We hope to satisfactorily answer all the concerns. We send you our revised manuscript (marked in yellow), and the response to your comments detailed point-by-point.
Reviewer 2:
Comments and Suggestions for Authors
Please include a separate "limitations" section in the manuscript.
Answer: A separate “limitations” section was included at the end of the Discussion section (marked in yellow).
Comments on the Quality of English Language: still needs improvement
Answer: The English language was reviewed again in the whole text.
Sincerely,
Dr. Rebeca López-Marure
Corresponding author